# The Innate Antiviral Response in Animals: An Evolutionary Perspective from Flagellates to Humans

**DOI:** 10.3390/v11080758

**Published:** 2019-08-16

**Authors:** Karim Majzoub, Florian Wrensch, Thomas F. Baumert

**Affiliations:** 1Inserm, U1110, Institut de Recherche sur les Maladies Virales et Hépatiques, Université de Strasbourg, 67000 Strasbourg, France; 2Institut Hospitalo-Universitaire, Pôle Hépato-digestif, Hôpitaux Universitaires de Strasbourg, 67000 Strasbourg, France; 3Institut Universitaire de France, 75231 Paris, France

**Keywords:** innate immunity, TLR, RLR, cGAS, STING, viral sensing, evolution, animals

## Abstract

Animal cells have evolved dedicated molecular systems for sensing and delivering a coordinated response to viral threats. Our understanding of these pathways is almost entirely defined by studies in humans or model organisms like mice, fruit flies and worms. However, new genomic and functional data from organisms such as sponges, anemones and mollusks are helping redefine our understanding of these immune systems and their evolution. In this review, we will discuss our current knowledge of the innate immune pathways involved in sensing, signaling and inducing genes to counter viral infections in vertebrate animals. We will then focus on some central conserved players of this response including Toll-like receptors (TLRs), RIG-I-like receptors (RLRs) and cGAS-STING, attempting to put their evolution into perspective. To conclude, we will reflect on the arms race that exists between viruses and their animal hosts, illustrated by the dynamic evolution and diversification of innate immune pathways. These concepts are not only important to understand virus-host interactions in general but may also be relevant for the development of novel curative approaches against human disease.

## 1. Introduction

The animal kingdom, including humans, has evolved while facing constant threats from viral elements. Viruses can be, in some cases, beneficial for a given animal species and drive its evolution [1]. However, their uncontrolled replication may cause disease and prove fatal to their hosts. Consequently, animal cells have evolved devoted pathways which (1) sense and recognize pathogen-associated molecular patterns (PAMPs) and, more particularly, virus-associated molecular signatures; (2) initiate signaling cascades stemming from the site of detection, translocating the information to the nucleus; and (3) induce a transcriptional program that confers an antiviral state to the host (Figure 1). Interestingly, a closer examination of individual factors constituting these pathways shows a different conservation status between different animal species. While genes encoding sensors and signaling platforms are generally well conserved amongst animals, virus-stimulated genes (VSGs) are clearly less so and are subject to faster evolution [2,3].

In vertebrates, one such VSG is the secreted interferon (IFN) cytokine, that signals in an autocrine and paracrine fashion. Secreted IFN molecules bind to cell-surface receptors and initiate signal transduction involving the Janus kinase/signal transducer and activator of transcription (JAK–STAT) pathway. This pathway induces the transcription of a major antiviral program composed of hundreds of so-called IFN-stimulated genes (ISGs) that comprise effectors of the cell-autonomous antiviral defense [4]. A lot of our understanding of the innate antiviral immune system in animals is a result of studies conducted in vertebrates and more particularly in mammalian species. Therefore, the IFN system has been heavily studied over the last 60 years. However, during the last two decades, we came to appreciate that the IFN system, as we know it, is a vertebrate particularity. Indeed, while some animal species like insects or nematodes are devoid of IFNs and rely on RNA interference (RNAi) as the major antiviral pathway, some others, like mollusks, have conserved all the components that lead to IFN production but have no obvious homologs of type I IFN cytokines. Nevertheless, these are predicted to use an IFN-like antiviral cytokine [5]. In fact, the IFN cytokine itself seems to be an evolutionary novelty, however, the pathways dictating its production existed early in metazoan evolution [3,6] (Figure 2). Interestingly, IFN is not the only VSG induced upon viral detection in mammals. Certain ISGs that directly interfere with the viral life cycle like Viperin are also immediately induced after viral infection in an IFN-independent fashion [7,8,9].

In this review, we will focus on the conserved pathways that are responsible for sensing viral PAMPs, signaling and inducing antiviral genes upon infection in animals. We will start by describing our current view on the immediate activation of the IFN and NF-*k*B pathways in vertebrate species. We will then zoom in on two important viral nucleic acid receptor families, Toll-like receptors (TLRs) and RIG-I-like receptors (RLRs), describe their function in viral RNA detection and their conservation across animal species. Next, we will focus on a central hub in the signaling pathways induced by DNA viruses, the stimulator of IFN genes (STING). We will then examine how the evolutionary conflicts between viruses and host immune factors are shaping antiviral immunity in animals.

## 2. Anti-Sense Targeting, an Ancestral Antiviral System

Viruses and transposable elements are very powerful drivers of evolution [1,19]. However, their uncontrolled replication and spread can be catastrophic to host cells. It is therefore not surprising that every known living species on the planet has evolved measures to recognize and counteract parasitic genetic elements. It is suggested that anti-sense mediated targeting of viral nucleic acids was the most primordial strategy protocells used to fend off viral threats [18]. This is illustrated by Argonaute and CRISPR-based defenses in bacteria, archaea and RNAi systems in plants, all of which rely on anti-sense nucleic acids that program a nuclease to target and degrade the complementary invading viral genome [20,21,22]. Because many of the core RNAi machinery components can be found in all eukaryotic superkingdoms, it is thought that this antiviral defense mechanism predated the emergence of pattern-recognition receptor (PRR)-based immunity (Figure 2) [23]. Interestingly, antiviral defense in eukaryotes has diversified greatly during evolution, with some species maintaining RNAi-based defenses [24] and others innovating and adopting completely novel antiviral strategies.

In chordates and more particularly in vertebrate animals, the emergence of IFN-I and a recombinational adaptive immune system seems to coincide with the loss of RNAi as the main antiviral mechanism in somatic cells. There is strong evidence of an intrinsic incompatibility between an antiviral RNAi and the PRR-IFN system. For instance, while long dsRNAs can produce functional small interfering RNAs (siRNAs) in stem cells in the absence of IFN, the same dsRNA molecule is not processed onto siRNA and is sensed as a PAMP in vertebrate somatic cells [25,26]. Interestingly, experimental evidence of antiviral RNAi in mammals seems to be limited to specialized pluripotent cells in which the PRR-IFN system is not fully deployed yet. The emergence of both the IFN-I (innate) and somatic DNA recombination systems (adaptive) in vertebrates constituted a major evolutionary event that dispensed them from using RNAi for antiviral purposes. Interestingly, retrotransposition and selfish transposable elements were determinants in the acquisition of these two systems [3,27,28,29].

## 3. Vertebrate Antiviral Immunity

The emergence of somatic DNA recombination in vertebrate animals was considered an “immunological Big Bang” [28,30]. Indeed, somatic DNA recombination in specialized B and T cell lineages provided jawed vertebrates with large repertoires of major histocompatibility complexes (MHC), T-cell receptors (TCRs) and immunoglobulins (Igs). Until relatively recently, adaptive immunity was believed to be exclusive to gnathostomes (jawed vertebrates). We now know that agnathans (jawless vertebrates), including lampreys and hagfish, have also evolved an equivalent adaptive immune system with specialized lymphocytes termed, VLRA, VLRB and VLRC cells, in which specific variable lymphocyte receptors (VLRs) are produced through somatic leucine-rich repeat (LRR) rearrangements [30]. In both gnathostomes and agnathans, somatic recombination events in specialized cells permitted a pathogen-tailored response and endowed vertebrate species with an immune memory.

Until the end of the last century, most vertebrate immunologists concentrated their efforts on studying the adaptive arm of the immune system. Nearly 30 years ago, Charles Janeway predicted the presence of an evolutionary ancient immune system that detects conserved microbial and danger signals, termed the pathogen-associated molecular patterns (PAMPs) and danger-associated molecular patterns (DAMPs), respectively. Janeway predicted that this innate immune system precedes and instructs the adaptive system [31,32] and posited that PAMPs and DAMPs must be sensed by germ-line encoded PRRs. At that time, the innate immune components were severely understudied and TLRs, RLRs and STING’s respective functions in immunity were completely unknown. This illustrates the immense leap forward the innate immunity field has experienced in the last three decades.

Today, we know that vertebrates share with other invertebrate animals specialized phagocytic cells that are able to discriminate between self and non-self. By recognizing general pathogen molecular patterns (PAMPs, e.g., viral double-stranded RNA) and danger signals (DAMPs), these cells establish an immediate and general inflammatory response translated into an antimicrobial and/or antiviral state. Although mainly studied in vertebrates, the machineries responsible for these responses transcend this group and are fairly conserved in all animals. We will describe a particular arm of the innate immune pathways, the innate antiviral system that is best studied in mammals.

## 4. Detecting Viral Patterns

Unlike bacteria, viruses represent a unique challenge for PRRs because they possess few unique signatures that could serve as PAMPs [33]. However, viral nucleic acids (DNA or RNA) could have peculiar biochemical features that differentiate them from endogenous host RNA [34]. In RNA molecules, for example, the lack of a 7-methylguanosine cap structure, double strandedness or the tri- or bi-phosphorylation at their ends are often used by PRRs for self/non-self-discrimination. PRRs that detect viral infection can be classified into four families: TLRs, RLRs, AIM2-like receptors (ALRs) and the cGAS-STING sensors [35]. After viral detection, PRR-mediated signaling directly or indirectly induces transcription factors, including IFN-regulatory factors (IRFs) and nuclear factor *k*-B (NF-*k*B) to upregulate expression of VSGs including pro-inflammatory cytokines. Another class of PRRs such as double-stranded RNA (dsRNA) activated protein kinase R (PKR; also known as eIF2AK2), adenosine deaminase acting on RNA 1 (ADAR1) and 2′-5′-oligoadenylate synthetase 1 (OAS1) also contribute to innate immunity [34]. These also recognize viral signatures; however, their main function is not necessarily to induce a transcriptional immune response, but rather to directly attack viral RNA by degrading it or inhibiting its translation. For this reason, these are not usually considered receptors.

Viruses are obligatory intracellular parasites, therefore their detection by PRRs most often occurs in the intracellular milieu. Endosomal transmembrane TLRs, including TLR3, TLR7 and TLR8 recognize dsRNA in the endosome lumen [36,37,38]. RLRs including RIG-I [39], melanoma differentiation associated gene 5 (MDA5) [40], and laboratory of genetics and physiology 2 (LGP2) [41,42] detect viral RNAs in the cytosol, whereas cytosolic viral DNA is mainly recognized by cyclic-GMP-AMP (cGAMP) synthase (cGAS) [43]. Therefore, the nature of the viral particle (e.g., enveloped vs. non-enveloped) and the viral genome (e.g., DNA vs. RNA) dictates which of these receptors recognizes the infection first (Figure 1).

AIM2 (absent in melanoma 2) can also recognize viral DNA just in some immune cells [44]. TLRs are predominantly expressed in immune cells, like dendritic cells (DCs), macrophages and B cells; in contrast, the major cytosolic RNA and DNA sensors, RLRs and cGAS, are expressed at varying levels in most cell types [45].

### 4.1. RNA Viruses

Single-stranded RNA (ssRNA) is a potent TLR7 and TLR8 ligand, while TLR3 is specific for dsRNA. TLR3, for example, recognizes dsRNA viruses from Reoviruses [36], but can probably also recognize dsRNA intermediates from (+) strand RNA viruses like coxsackievirus and West Nile virus (WNV) and (-) strand RNA viruses like the human respiratory syncytial virus (HRSV) [46]. Indeed, all RNA viruses are thought to produce dsRNA intermediates as part of their replication cycle, so both ssRNA and dsRNA viruses have the potential to be sensed by TLR3. TLR7 and 8, on the other hand, have been shown to prefer ssRNA ligands from (-) strand RNA viruses such as vesicular stomatitis virus (VSV) and influenza A virus (IAV) [37,47]. When it comes to RLRs, most RNA viruses have been shown to be detected by RIG-I or MDA5, as these receptors have a high affinity to dsRNAs. While RIG-I prefers viral RNAs bearing di- or tri-phosphate groups at their 5′ end, MDA5 preferentially binds longer dsRNA ligands [48,49,50,51]. Consequently, some viruses like dengue virus (DENV) or WNV can be detected by both MDA5 and RIG-I [52,53,54], while other viruses such as Enterovirus 71 (EV71) or encephalomyocarditis virus (EMCV) are mainly detected by MDA5 [55,56]. HRSV, influenza A/B, VSV, hepatitis C virus (HCV) and Newcastle disease virus are preferentially recognized by RIG-I [50,52]. The last member of the RLR family, LGP2 has RNA binding but no signaling domains and is thought to regulate RIG-I and MDA5 signaling. For instance, LGP2 has been shown to contribute to MDA5-mediated responses to mengovirus and EMCV [57,58,59]. In some cases, DNA substrates have also been shown to be able to activate the RLR pathway through polymerase III-transcribed RNA [60,61,62].

### 4.2. DNA Viruses

While the cytosolic recognition of viral RNA is almost exclusively mediated by RLRs, several proteins have been proposed to play a role in DNA sensing and triggering innate immune responses, such as the DNA-dependent activator of IFN-regulatory factors (DAI), DDX41, RNA polymerase III, IFI16 and DNA-PK [62,63,64,65,66,67]. However, among all the proposed sensors, only cGAS knock-outs can completely shut down IFN production in response to cytosolic DNA [43]. The cGAS protein is now thought to be the major viral DNA sensor and has been shown to detect adenovirus, human papillomavirus (HPV), herpes simplex virus-1 (HSV-1) and cytomegalovirus (CMV) [43,68,69,70]. AIM2 has also been shown to activate the inflammasome upon DNA stimulation [71] but will not be discussed in this review.

## 5. Sending the Message to the Nucleus

### 5.1. TLR Signaling

RNA ligands cause the endosomal transmembrane TLR3, 7 and 8 to dimerize and then to oligomerize through their cytoplasmic TIR (Toll/IL-1 receptor) domains. This allows TLRs to recruit signaling adaptors via TIR–TIR interactions [35]. TLR3 recruits the adaptor protein TRIF (TIR-domain-containing adapter-inducing IFN-β) [72,73]. TRIF is able to play a dual role by inducing the IFN or the NF-*k*B pathways. When it comes to IFN, after activation, TRIF recruits the ubiquitin ligase TRAF3 (tumor necrosis factor receptor–associated factor 3) through an ubiquitination mechanism which in turn recruits TANK-binding kinase 1 (TBK1) [74,75]. The TRIF/TBK1 complex is then able to phosphorylate the transcription factor IRF3, triggering its dimerization and nuclear translocation. Phosphorylated IRF3 dimers specifically bind to IFN-stimulated response elements (ISREs) present in the IFN-β gene promoter which leads to the transcription of this cytokine [76]. TRIF can also recruit RIPK1 (receptor-interacting serine/threonine-protein kinase 1) that leads to the activation of the IKK complex, releasing the NF-kB transcription factor from its I*k*B inhibitory subunit and resulting in its translocation to the nucleus to induce the transcription of pro-inflammatory cytokines [17] (Figure 1).

Unlike TLR3, the activation of TLR7 and TLR8 recruits the adaptor protein MyD88 (myeloid differentiation primary response 88) through TIR–TIR domain interaction. MyD88 death domains oligomerize which triggers the formation of the Myddosome signaling complex consisting of MyD88 and the IRAK family of kinases (IL-1 receptor–associated kinases), IRAK 1, 2 and 4. Through a series of phosphorylations and the help of the E3 ubiquitin ligase TRAF6, the Myddosome is able to recruit and activate the transcription factors IRF7, IRF5 and NF-kB that translocate to the nucleus to induce the transcription of IFN-α genes and other proinflammatory cytokines [77,78,79,80,81,82,83] (Figure 1).

### 5.2. RLR Signaling

RLRs (RIG-I, MDA5 and LGP2) are characterized by a central DEAD-box helicase/ATPase domain and a C-terminal regulatory domain (CTD) essential for RNA recognition and autorepression in the absence of RNA ligands. With the exception of LGP2, RLRs also possess two N-terminal caspase activation and recruitment domains (CARDs). Upon RNA binding RIG-I is remodeled into an active conformation in which the CTD and helicase domains organize into a ring around the RNA ligand and the CARD domains are exposed [84,85] which facilitate their interactions with other CARD domains resulting in RIG-I tetramers. Although RIG-I and MDA5 share similar domain architectures, MDA5 seems to prefer longer dsRNA, assembling along these molecules to form helical, filamentous oligomers [86,87]. A poly-ubiquitination reaction by ubiquitin ligases like Riplet and TRIM25 (tripartite motif-containing 25), is thought to enhance RIG-I and MDA5 oligomerization and activation [88,89,90,91].

RIG-I and MDA5 oligomers then serve as a scaffold for binding to the adaptor protein MAVS (mitochondrial antiviral signaling protein, also known as IPS-1, VISA, and CARDIF) [92,93,94]. MAVS has been shown to be critical for mounting an efficient immune response to infection by several RNA viruses [95]. Its C-terminal transmembrane domain is inserted into the outer mitochondrial membrane [96], whereas its N-terminal CARD domain mediates its aggregation on the mitochondrial surface by interacting with the tandem CARDs of RIG-I or MDA5 oligomers [97,98]. MAVS aggregates then recruit several E3 ubiquitin ligases including TRAF2, TRAF5 and TRAF6. Although TRAF-mediated ubiquitination is essential to activate MAVS downstream signaling, the ubiquitination targets of TRAF remain unknown [99]. Subsequently, the ubiquitin sensor NEMO (NF-*κ*B essential modulator, also known as IKKγ) [100,101] is then recruited to the MAVS/TRAFs complex, which in turn recruits IKK and TBK1 to the MAVS complex leading to activation of NF-*k*B and IRF3 and their translocation to the nucleus to induce the transcription of antiviral genes [99,100,101,102] (Figure 1).

### 5.3. The cGAS-STING Axis

After TRIF and MAVS were discovered, STING was identified as a third adaptor protein that is also able to activate IRF3 and IFN production [103,104]. STING is an endoplasmic reticulum (ER) resident membrane protein with cytoplasmic C- and N-termini. STING has been shown to be essential for DNA-mediated IFN production in different tissues, for example, it is crucial for host defense against the DNA virus HSV-1 [105]. STING has also been shown to sense cyclic dinucleotides (CDNs), which are the second messengers known to be produced by bacteria such as *Listeria monocytogenes* [106,107,108,109]. Although it can bind bacterial CDNs, STING is unable to bind DNA and relies on an upstream sensor, cGAS [43]. cGAS is an enzyme that contains a nucleotidyltransferase (NTase) domain and can synthesize the second messenger 2′3′-cyclic GMP-AMP (cGAMP) from ATP and GTP upon DNA recognition (Figure 1). Loss of cGAS in various cell lines and also in vivo results in a complete loss of type I IFN induction upon DNA delivery or viral infections [110,111]. cGAS preferentially binds longer DNA (>45 bp) as a dimer to form stable protein-DNA ladder networks responsible for strong cGAMP production [112,113]. A unique cGAMP isomer termed 2′3′-cGAMP with particular phosphodiester linkages is produced by cGAS [114,115]. 2′3′-cGAMP is a potent STING ligand and has a higher affinity to this protein than other cGAMP molecules containing different phosphodiester linkages such as 2′2′-cGAMP, 3′2′-cGAMP or bacterial CDNs [70,115]. Apart from activating STING in the cell where cGAS initially detects viral DNA, cGAMP second messengers can also travel to neighboring cells, through gap-junctions [114] or after being packaged in newly formed virions [116,117]. This intercellular transfer of free or packaged cGAMP permits uninfected cells to mount a preventive IFN response, protecting them from infection or providing a faster response to DNA viruses that encode cGAS antagonists.

Upon cGAMP binding, STING undergoes a conformational change that results in the release of its C-terminal tail (CTT) from its autoinhibitory state and in the formation of STING homodimers that translocate to perinuclear regions to colocalize with TBK1 [105,118,119]. TBK1 recruitment results in the phosphorylation of STING and the phosphorylated site serves as a platform for IRF3 dimerization and activation which ultimately results in IFN- β induction [120] (Figure 1). STING has also been shown to induce NF-*k*B, MAP kinase and STAT6 activation, as well as the stimulation of LC3 puncta formation, a hallmark associated with autophagosome formation [119,121,122,123]. However, the molecular mechanisms by which STING induces these non-IFN responses remain poorly understood.

## 6. TLRs, an Ancient Family of Receptors

TLRs comprise an ancient family of membrane-spanning receptors that recognize ligands through their extracellular domains and initiate an intracellular response upon stimulation (see above). The *Toll* gene was first identified as a developmentally important gene in *Drosophila* in 1985 [124]. In the mid-1990s the discovery that this gene also plays an essential role in the ability of *Drosophila* to resist fungal infections connected for the first time Toll receptors to innate immunity [125,126]. Although in flies Toll functions as a cytokine receptor, a human Toll receptor (TLR4) was rapidly identified [127,128] and shown to induce an immune response in mice after induction by LPS [129]. We now know that there are ten TLRs in humans that can respond to many bacterial and viral PAMPs [130].

Prototypical TLRs contain three structural elements, a hydrophobic ectodomain containing a variable number of LRRs, a transmembrane domain and a TIR domain, which mediates downstream signaling through adaptor proteins [131]. TLRs are likely very ancient immune sentinels since two of their characteristic building blocks (LRR and TIR domains) are observed in placozoans (e.g., *Trichoplax* animals) [132] and Porifera (e.g., Sponges) [131]. Full TLRs were detected in Cnidarian species, like the starlet sea anemone (*Nematostella vectensis*; one single TLR) [133,134] and the acroporid corals (*Acropora digitifera*; four TLRs) [135] (Figure 2). Interestingly, both developmental and immunological roles of TLRs have been described in Cnidarians. TLRs from both the sea anemone (*Nematostella vectensis*) and the mountainous star coral (*Orbicella faveolata*) have been shown to signal via MyD88 leading to NF-*k*B activation [133,136].

In the Bilateria phylum, TLRs can be found in most studied species, however, their numbers vary greatly among species, ranging from a single TLR in Nematodes like *Caenorhabditis elegans*, to over two hundred in echinoderms like the pacific purple sea urchin *Strongylocentrotus purpuratus* (Figure 2). The expansion of the TLR repertoire in some animals like the sea urchin, reflects the adaptation of their immune arsenal to rapidly changing environmental stressors [137].

Amongst a multitude of other innate immune factors in this species, such as NACHT domain-LRRs and Scavenger receptors, sea urchin genomes encode for 222 TLRs. Among those, 211 TLRs belong to a greatly expanded set of genes with vertebrate like features, many of which seem to have duplicated recently. The high prevalence of pseudogenes (25% to 30%) among those might reflect a history of strong positive selective pressures.

Another phylum where TLRs have undergone a significant expansion is in Mollusca [138], like the pacific oyster *Crassotrea gigas* [139] (Figure 2). The Pacific Oyster encodes for 83 TLRs in total, potentially reflecting a highly specialized response to environmental challenges and response to pathogens. The spread of pathogens in *C. gigas* natural habitats occurs very quickly, which is highlighted by the mass mortality events the Ostreid Herpesvirus 1 (OsHV1) has caused in many oyster nurseries. TLR sensing of OSHV1 results in the differential regulation of more than a thousand genes, many of which are related to viral infection (e.g., cytosolic DNA sensing and DNA replication) [5,139].

In contrast to the very diverse set of TLR repertoires found in other Bilateria species (e.g., Nematodes, sea urchins and oysters), chordates and more particularly vertebrates contain roughly equal numbers of TLRs, reflecting the reduced need for highly diversified pattern recognition due to the acquisition of adaptive immune components (Figure 2). In general, vertebrate TLRs can be grouped into six major families [15]. The families responsible for sensing of viral PAMPs are the TLR3 family, which recognizes dsRNA, the TLR7 family (including TLRs 7, 8 and 9) which recognizes nucleic acid motifs and the large TLR11 family (TLR11, 12, 13, 19, 20, 21, 22, 23 and 26). The reduced number of TLRs in vertebrates does not necessarily mean that the TLR-response in those species cannot be tailored to a particular environment. A peculiar example is TLR22, one of two virus sensing TLRs present in the pufferfish *Takifugu rubripes*. TLR22 is widely conserved among teleosts and amphibians but does not seem to be present in avian or mammalian animals, which indicates that TLR22 might be required only in vertebrates living in water [140]. In mammals, one last case of TLR adaptation and rapid evolution that is worth mentioning comes from bat species. Analyses of TLR evolution in bats reveal adaptations acquired by TLRs 3, 7, 8 and 9, with unique mutations fixed in ligand-binding sites [141,142]. These adaptations are thought to stem from the unique lifestyle of bat species, that are the only known flying mammals, and that represent important viral reservoirs [143].

## 7. RLRs across Animal Species

Evolutionary studies paint a complex and dynamic picture of the emergence and functional diversification of RLRs across the animal kingdom. Initially, several studies proposed that RIG-I and MDA5/LGP2 evolved in animals independently through gene fusion and domain grafting events [16,144]. For instance, it has been proposed that the two CARD domains have been acquired by RIG-I and MDA5 in two separate events: The first domain being gained by the ancestor of RIG-I and MDA5 before their duplication and the second acquired after their divergence [16]. These studies suggested that full-length RLRs are a vertebrate-specific evolutionary novelty, although their building blocks may have been present in closely related invertebrate animals [16,144]. A more recent study challenges this view and finds that the RLR-based immunity is not vertebrate-specific but originated in the earliest multicellular animals [14] (Figure 2). In this study, the authors show that RLRs functionally diversified through a series of gene duplication events, followed by protein-coding changes that modulated their RNA-binding properties. Using homology-based gene prediction based on confirmed human RLRs the authors were able to identify full-length RLRs in early-branching animal genomes, including Porifera (e.g., sponges) and Cnidaria (e.g., jellyfish). However, they were unable to identify RLRs in non-metazoan eukaryotes, including fungi and choanoflagellates [14] (Figure 2). It is therefore proposed that the ancestral RLR (RIG-I/MDA5/LGP2anc) duplicated in Bilateria to give rise to RIG-I and MDA5/LGP2 lineages, followed by a more recent duplication of the MDA5/LGP2 ancestor, giving rise to MDA5 and LGP2 lineages in jawed vertebrates after their split from jawless vertebrates [14]. The emergence of RLRs early in animal evolution is a very plausible scenario, since other components of the signaling pathways downstream of RLRs, like the IRF genes, are also found in early metazoans [6] (Figure 2). Another recent evidence suggesting that RLRs predated vertebrate evolution comes from studies performed in mollusks (pacific oyster; *C. gigas*). The invertebrate *C. gigas* not only encodes up to 12 RLRs, but also MAVS, TRAF6, TBK1 and IRF family proteins, which have been shown to have functional antiviral roles [5,139,145] (Figure 2).

Even though there is no consensus on the exact evolutionary history of RLRs, it is clear that these receptors (and/or their building blocks) existed very early in metazoan evolution and most importantly, they are subject to a very dynamic evolution. This is illustrated by the lineage-specific loss of RLR genes in many species. For example, although MDA5 and LGP2 homologs were found in many teleost fish, RIG-I homologs have only been identified in some fish species like salmon and carp [146]. RIG-I is absent in the chicken genome although MDA5 and LGP2 are both present [147,148]. Interestingly, chickens suffer severely from avian influenza virus (AIV) infection compared to ducks (that do possess the RIG-I gene) which could be due to the loss of RIG-I affecting their first line of defense in epithelial cells [148]. Most studied mammals possess RIG-I, however, it has been lost in at least one mammalian species; the Chinese tree shrew [149]. Interestingly, with the loss of RIG-I, both MDA5 and LGP2 have undergone strong positive selection in Chinese tree shrews, and positively selected sites in MDA5 endowed the substitute function for the lost RIG-I [150]. Another eloquent example illustrating the dynamic evolution of these receptors is the loss of all RLR genes (RIG-I, MDA5 and LGP2) in insects (Figure 2). In *Drosophila*, for example, although the NF-*k*b and JAK/STAT pathways are present and contribute to antiviral defenses [151,152], all components of the RLR-MAVS-IRF-axis have been lost. Instead, *Drosophila* like other insects and relies on the RNAi mechanism as the major antiviral system protecting it from viral infections [24,153,154]. Interestingly the RNase III Dicer-2, a central player in insect antiviral immunity, responsible for generating small interfering RNAs (siRNAs), also contains an N-terminal DExD/H-box helicase domain that is highly homologous to the helicase domains of vertebrate RLRs [155,156]. Moreover, Dicer-2 has been shown to be responsible for the transcriptional upregulation of an antiviral gene (Vago) that could function as a cytokine by activating the JAK/STAT pathway and triggering systemic antiviral immunity in various mosquito tissues [157,158]. Although the pathway leading to the transcriptional activation of Vago is still poorly understood in insects, these studies established that DExD/H-box helicase containing proteins, like Dicer and RLRs, may represent an evolutionarily conserved set of viral nucleic acid sensors that direct antiviral responses in animals [159]. One last observation exemplifying the dynamic and rapid evolution of these receptors comes from mammalian species. Indeed, RLRs seem to be experiencing very recent adaptive changes in some mammals. For example, RIG-I seems to have accumulated adaptive changes altering its RNA-binding properties throughout mammalian evolution [160]. Moreover, in humans, for example, a number of protein-coding polymorphisms have been identified in RIG-I which may contribute to differences in viral susceptibility and risk of autoimmune diseases [14,161,162].

## 8. The cGAS-STING Pathway, a New–Old Axis

STING presence in animal genomes is probably more ancient than that of RLRs, since STING homologs can be found in most animal phyla including unicellular choanoflagellates (Figure 2) [12,163]. Furthermore, the ability of STING to bind CDNs seems to be an ancient property. In an elegant study, Kranzusch and colleagues show that a STING homolog in the starlet sea anemone *N. vectensis (*nvSTING) is not only structurally very similar to that of human STING but is also able to bind 2′3′cGAMP with very high affinity [13]. However, STING’s CTT domain, which is crucial for TBK1 recruitment and downstream IFN induction, appeared only in vertebrate species [164]. Consequently, nvSTING lacking the CTT is unable to induce IFN-β production in response to CDNs when transfected in mammalian cells [13]. The lack of a CTT domain in invertebrates does not mean that STING could not have an immune function in these animals. A first indication comes from invertebrate species like the Lophotrochozoa phylum that includes the pacific oyster *C. gigas* and the annelid worm *Capitella teleta.* In these animals, an unusual STING architecture can be found, where a STING domain is fused to a TIR domain, known to be involved in innate immune signaling [164,165]. The second indication that STING lacking a CTT could function in immunity comes from arthropods. Recent studies in *Drosophila*, that lack an IFN system, clearly show that STING is important for antimicrobial and antiviral NF-*k*B activation in this model [151,166] (Figure 2). Interestingly, the emergence of the CTT domain of STING in vertebrate species seems to coincide with the development of the IFN system. Nevertheless, STING CTT domain function, which dictates downstream signaling, seems to be plastic amongst vertebrate species. In a recent study, authors show that STING CTT-dependent activation of IRF3 and NF-*k*B varies between vertebrate species [167]. While STING CTT from mammalian species is able to induce a strong IFN-β and a weaker NF-*k*B response, an extension of this domain in ray-finned fish species elicits a dramatic enhancement of NF-*k*B activation and weaker IRF3-IFN signaling [167]. Another indication of STING CTT structure-function plasticity comes from bat species. A highly conserved and functionally important serine residue (S358) in STING’s CTT domain is lost in bats [168]. The replacement of this critical residue in this mammalian species significantly dampens STING-dependent IFN activation. The authors of this study suggest that the lifestyle of bat species (e.g., flight induced cytosolic DNA, high viral titers) imposes a strong selective pressure on STING. This results in functionally dampened sensing and signaling mechanisms to avoid IFN overactivation and to cope with high cytosolic DNA content. Taken together, present studies suggest an evolutionarily ancient role of STING in antiviral immunity and modulation of its structure and function to accommodate species-specific pathogen burdens.

The picture is less clear for the cGAS enzyme when it comes to antiviral immunity. Although cGAS homologs have been identified in a variety of ancient metazoan lineages [12,163], it is believed that the ability of cGAS to bind and detect dsDNA emerged in vertebrates. Indeed, cGAS’ zinc-ribbon domain, required for DNA binding and cGAMP synthesis in response to DNA in the cytosol, seems to be a vertebrate innovation [169,170,171]. Interestingly, primate cGAS seems to have undergone rapid evolution in this lineage, as observed by the positive selection at its nucleic acid binding interfaces [172]. These studies argue that although the cGAS enzyme existed early in metazoans, its function has been repurposed for DNA sensing only recently in vertebrates. Clearly, cGAS and STING seem to have acquired novel features throughout evolution. Specifically in vertebrates cGAS evolved the zinc ribbon motif to detect DNA and STING evolved the CTT domain that expanded its signaling potential.

## 9. Defenses and Counter Defenses

As obligate intracellular parasites, viruses have evolved an array of evasion mechanisms to escape their elimination by the host’s immune system. Interestingly, viral antagonism is a general strategy and is not a peculiarity of animal viruses. Many bacteriophages, for instance, encode CRISPR-Cas inhibitors, termed anti-CRISPRs, to counter prokaryotic antiviral systems [173]. Plant viruses also encode viral suppressors of RNAi (VSRs) the main antiviral system in plant cells [20]. 

Likewise, several evasion strategies and immune antagonisms by animal viruses have been described [174,175,176,177,178]. These include hiding the viral genome from immune detection, shutting off host translation or transcription machineries, inhibiting host RNA processing and trafficking and interfering directly with either proteins that sense viral presence, or factors that signal the information to the nucleus. Since interfering with the innate immune system is less damaging for the host than targeting vital cellular machineries (e.g., translation), many studied viruses seem to have opted for this strategy. Several studies describe viral evasion mechanisms at both the recognition and sensing step (TLRs, RLRs and cGAS-STING) or at the downstream signaling steps through the targeting of proteins such as MAVS, TBK1, IRF3, IRF7 and NF-*κ*B. Evasion strategies and immune antagonisms by animal viruses are a very active area of research, that have yielded a rich literature in the past few years. We will here just give some select examples of viral strategies that curb sensing and signaling by TLRs, RLRs and cGAS-STING in animals, with an obvious bias towards viruses infecting humans. For a more complete picture on the subject, readers can refer to excellent reviews, published recently, describing those strategies [174,175,176,177,178].

### 9.1. TLR Evasion Strategies

TLR signaling has been shown to be inhibited by the vaccinia virus (VACV) protein A46R, that targets specific TIR-domain-containing adaptor proteins. A46R itself contains a TIR domain which allows it to competitively interact with TIR-domain-containing complexes such as Myd88, TRIF or TRAM, thereby inhibiting the activation of both NFkB and IRFs [179,180]. Human T-cell leukemia virus type-1 (HTLV-1) is also able to interfere with TLR4-dependent signaling. The HTLV-1 encoded viral protein p30 binds and disables a transcription factor, PU.1, required for TLR4 surface expression [181]. TRIF, an important player in the TLR signaling cascade, is a target of choice of many viruses. The NS3/4A protease of HCV and the 3C proteases of several picornaviruses such as coxsackievirus B, EV71 and hepatitis A virus (HAV), can all recognize and proteolytically cleave TRIF, producing TRIF fragments that are unable to signal [182,183,184,185,186].

### 9.2. RLR Subversion by Viruses

When it comes to RLRs, one basic strategy used by cytosolic viruses to escape surveillance is to simply prevent these receptors from accessing viral genomes. DENV, for example, replicates in convoluted membranes of the ER concealing its dsRNA intermediates from the cytosol and thereby prevents the activation of RLRs [187]. Other viruses like Ebola virus (EBOV) and Marburg viruses encode viral protein 35 (VP35) that tightly binds and ‘shields’ the viral genome from detection by RIG-I [188,189]. Another strategy used by viruses to ‘hide’ from RLRs consists of modifying the very molecular features these receptors rely on to recognize viral genomes. For example, both, Hantaan viruses from the Bunyaviridae family and Borna disease virus (BDV) from the Bornaviridae family, encode phosphatases that process the triphosphate group at their 5′ genome termini, to a 5′-monophosphate to escape RIG-I surveillance [190,191]. Lassa Virus (LASV) from the Arenaviridae family evolved a unique strategy in which its nucleoprotein (NP) acquired a 3′-5′ exonuclease activity, that enables it to digest free dsRNA, preventing the activation of RIG-I [192].

However, the most direct way of interfering with RLR function and their signaling partners is to either directly target them for cleavage and degradation or to manipulate their phosphorylation and ubiquitination statuses, which are crucial for their activation. Indeed, many viruses encode proteases that directly cleave RLRs. While the 3Cpro proteases of both poliovirus and EV71 cleave RIG-I, the 2Apro of EV71 cleaves MDA5 [193,194]. MAVS, a crucial hub for both RIG-I and MDA5-mediated signaling is also frequently targeted and cleaved by numerous viral proteases, such as 3Cpro from HAV, 2Apro from EV71, NS3–NS4A from HCV, 2Apro and 3Cpro from rhinovirus and 3Cpro from coxsackievirus B3 (CVB3) [184,185,194,195,196]. MAVS can also be indirectly degraded by particular viruses. For instance, measles virus (MeV) can trigger a selective form of autophagy, called mitophagy, responsible for the degradation of mitochondria, which leads to a decrease of MAVS abundance [197]. Another example of indirect MAVS degradation comes from studies with severe acute respiratory syndrome (SARS)-associated coronavirus (SARS-CoV). This virus has evolved a strategy in which its 9b protein localizes to mitochondria and subverts the cellular E3 ubiquitin ligase atrophin-1-interacting protein 4 (AIP4) to degrade MAVS [198].

Post-translational modifications of both MAVS and RLRs have also been shown to be subverted by viruses to inhibit their downstream signaling. NS1 proteins from many influenza A virus strains (IAV) interact with the host ubiquitin ligase TRIM25 and inhibit its oligomerization, a crucial step for its enzymatic activity of attaching Lys63-linked polyubiquitin to the CARD domains of RIG-I [24,199]. Other viruses encode deubiquitinating enzymes (DUBs) to remove the Lys63-linked ubiquitination off RIG-I. ORF64 from Kaposi’s sarcoma herpesvirus (KSHV), papain-like protease (PLP) from SARS-CoV, leader proteinase (Lpro) from foot-and-mouth disease virus (FMDV) and the ovarian tumor (OTU)-type proteins of arteriviruses and nairoviruses have all been shown to possess a deubiquitination activity and interfere with RIG-I mediated signaling [177,200,201,202].

RIG-I and MDA5 phosphorylation status can also be subverted by viruses. In normal conditions, phosphorylation of serine or threonine residues keeps RIG-I and MDA5 in an inactive state. Upon viral infection, PP1 phosphatases are recruited to dephosphorylate specific marks on those receptors and activate them. V proteins from Measles and Nipah viruses (MeV and NiV) act as decoys and have been shown to bind PP1-α and PP1-γ, sequestering them away from MDA5 and RIG-I [203,204].

### 9.3. Breaking Free from the cGAS-STING Axis

Similar to evasion strategies that counter the RNA sensing machinery described earlier, DNA viruses use numerous strategies to escape cGAS-STING-dependent detection and signaling. They could either hide their viral genomes or cleave, degrade, post-translationally modify or even relocalize DNA sensing and signaling factors [205]. Hepatitis B Virus (HBV), that causes chronic hepatitis and increases the risk of developing liver cirrhosis and hepatocellular carcinoma, has developed an array of mechanisms to inhibit the host’s immune systems (reviewed in [206]). Notably, the HBV polymerase can bind to STING to block its Lys63-linked ubiquitination, inhibiting the production of IFN-β [207]. Moreover, even though cGAS is expressed in human hepatocytes and is able to sense and signal upon transfection of naked relaxed-circular HBV DNA; during a natural infection, HBV DNA seems to escape cGAS detection, likely due to packaging of the genome into the viral capsid [208]. KSHV, another DNA virus has been shown to act on both cGAS and STING. Several KSHV proteins (e.g., ORF52 and LANA) can either sequestrate stimulatory DNA or directly bind to cGAS inhibiting its enzymatic activity [209,210]. KSHV has been also shown to encode a viral interferon regulatory factor (vIRF1) that interacts with STING thereby preventing TBK1 binding and STING activation by TBK1-dependent phosphorylation [211].

The NS3 protease of DENV, together with its NS2B co-factor, has been shown to target the residues 93–96 (LRRG) of human STING, leading to its cleavage and degradation [212,213]. Interestingly, mouse STING lacks these LRRG residues, and NS2B/NS3 of DENV is neither able to cleave the murine STING, nor to block murine IFN-β production. Therefore, it has been proposed that the inability of DENV to cleave mouse STING might explain its host tropism, as murine cells are not very susceptible to DENV infection [212,213].

## 10. Concluding Remarks

In animals, PRRs and their associated signaling pathways are early and potent cellular sensors of viral elements, that mobilize the organism’s defenses by inducing an antiviral state. Major advances have been made in the last two decades in the understanding of their function in mammalian immunity. New genomics data and gene editing tools can now let us interrogate PRR-like pathways in poorly studied animal species and define their evolutionary trajectories. Studying the evolution of immune components and their interplay with viral pathogens is extremely important since our immune responses to contemporary viruses have been shaped by our evolutionary responses to previous infections. The modern innate immune system is generally not yet optimized against modern viruses but rather was selected for by previous rounds of co-evolution with ancient viruses [214]. Studying the biological arms race between host and virus, referred to as the “Red Queen hypothesis” [215], in which each entity maintains a relatively constant fitness cost, will be instrumental in the fight against future infections. Such studies will help us understand many aspects of viral infections including viral zoonoses, tropism, global epidemics and disease progression. Furthermore, exploring these pathways and mechanisms for therapeutic purposes may offer novel strategies to cure human disease. Indeed, modulating the action of the aforementioned immune sensors is proving to be an effective strategy to develop vaccines and vaccine adjuvants [216,217,218,219,220] or to treat viral infections [221,222,223,224,225,226]. Finally, the use of TLR, RLR and STING modulators, to treat inflammation, auto-immune disease [227,228] and also in cancer immunotherapy [229,230,231,232,233,234,235,236] provides an eloquent incentive to continue studying these pathways and to look ahead with great optimism.

## Figures and Tables

**Figure 1 viruses-11-00758-f001:**
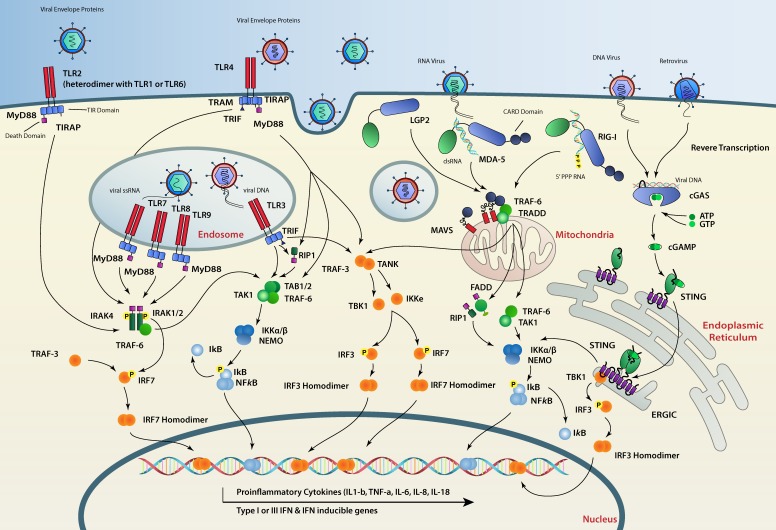
Detection of viral patterns by pattern recognition receptors (PRRs) and the signaling cascade in vertebrates. Viral patterns such as envelope proteins, viral RNA or viral DNA are recognized by host cell receptors, including Toll-like receptors (TLR) 3, 7, 8 and 9, RIG-I-like receptors (RLRs) RIG-I, MDA-5 and cGAS-STING. Activation of PRRs induces a signaling cascade which ultimately results in the phosphorylation and dimerization of IRF3 or 7 or the release of the inhibitory protein I*K*b from NF*k*B which then migrate to the nucleus and induce the expression of secreted interferon (IFNs), cytokines and IFN-stimulated genes (ISGs). Arrows indicate interactions and/or protein movements.

**Figure 2 viruses-11-00758-f002:**
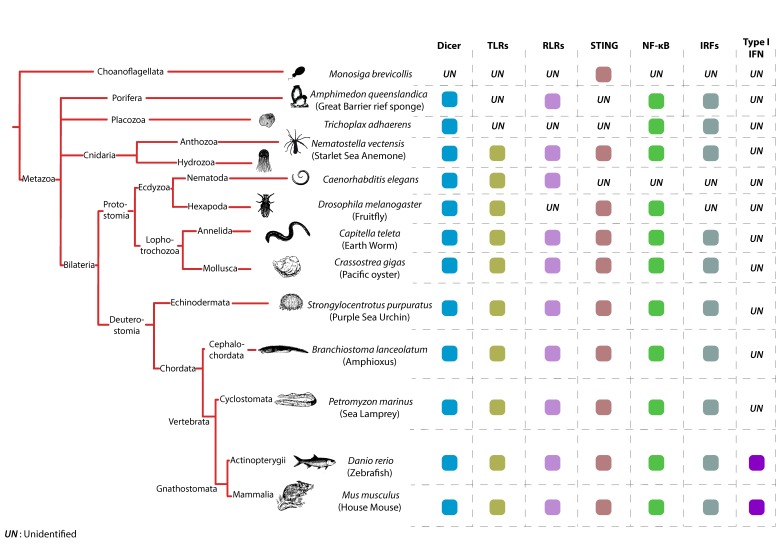
Conservation of key antiviral genes across the animal kingdom. Displayed is a simplified taxonomic branching order diagram of Metazoa with one outgroup (Choanoflagellata), adapted from [6]. Depicted is the presence of selected innate immune factors in representative species from Metazoa and Bilateria starting from Porifera up until Mammalia. The presence of DICER—a central component of the RNAi pathway, of the pattern recognition receptors TLRs and RLRs, of the signaling molecule stimulator of IFN genes (STING) and the effectors NF-kB, IRFs and IFN—is indicated by colored squares. Absence is indicated by UN (unidentified). References used to construct this figure are listed [3,6,10,11,12,13,14,15,16,17,18].

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
