# Peer review of "The Innate Antiviral Response in Animals: An Evolutionary Perspective from Flagellates to Humans"

_viruses, 2019, doi:10.3390/v11080758_

Round 1

Reviewer 1 Report

In the manuscript, the authors provide an overview of the various mechanisms by which animal cells sense and respond to viral pathogens. The authors do a satisfactory job at discussing the main components of innate immune sensing and signal transduction.

Where I found that the manuscript fell short is in integrating the description of the sensing/signaling components into a coherent discussion of the greater evolutionary context of their evolution. Because of this, there was a disconnect between the title and abstract, which seems to promise an integrated evolutionary discussion of innate immunity, and the body of the text which in many places has a strong bent towards mammalian systems/viruses. The manuscript would benefit greatly if the evolutionary picture of the systems being discussed were better integrated into the text.

Major Comments

'Rnai v/s PRR-based antiviral immunity' (Lines 395-424): As noted in the manuscript, the evolution of type I IFN appears to have been a defining split in vertebrate evolution. Many/most of the audience of this review will be familiar with IFN-I and its effectors, but will be less familiar with innate immunity in organisms lacking this core component of the mammalian innate immune response. Thus, in my opinion, it would make sense to elucidate this crucial difference very early on in the manuscript – perhaps as part of an expanded 'Vertebrate antiviral immunity' section.

'Defense & counter defenses' section (Lines 426-430): It would be nice to provide some specificity to the discussion of strategies viruses use in evading the innate immune response. The vast majority of viruses encode one or more proteins whose function it is to counteract or modify the activity of key proteins involved in the innate immune response. Whole reviews have been written on this topic, so clearly it would be impossible for the authors to cover this in depth. However, as written the audience is left in the dark as to how viruses are able to side-step the innate immune response to successfully replicate in their hosts.

Minor Comments

Lines 16/17: Humans should be included in the list of organisms from which our knowledge of innate immune system is derived.

Line 32: This sentence seems to overstate the frequency of virus-host interactions that are beneficial to the host.

Line 50: I would suggest replacing 'humans' with 'vertebrates', because, as noted later in the paragraph, humans are not the only species that utilize IFN.

Lines 64/65: This sentence is awkwardly worded.

Figure 2: As a means of providing a more holistic overview of immune system evolution, it would be nice to include columns for specialized phagocytic cells and adaptive immunity.

Lines 86-88: The adaptive immune system described here applies to vertebrate species from jawed fish to mammals. However, a similar adaptive immune system independently evolved in jawless fish. It would be nice to note this and highlight how much of the underlying signaling is conserved between T cells and VLR-A cells as well as B cells and VLR-B cells.

Line 99: The acronyms PAMP and DAMP should be properly defined (pathogen-associated molecular pattern and danger-associated molecular pattern, respectively).

'RNA viruses' section (Lines 133-149): It would be nice to note somewhere in this section that all RNA viruses produce dsRNAs as part of their replication cycle, so both ssRNA and dsRNA viruses have the potential to be sensed by MDA5/TLR3.

Line 258: Rephrase as 'all over the board' does not have any biological/evolutionary meaning.

Lines 266-267: Genes with a history of strong positive selective pressures often have expanded copy numbers and a high prevalence of pseudogenes (see e.g. olfactory receptors and certain tripartite motif-containing genes).

Lines 267-269: It is unclear what the authors are proposing with this sentence.

Author Response

We would like to thank the reviewers for their thorough reading of the manuscript. Their comments were very insightful and we took into account all their suggestions in the revised manuscript. Please find below a point-by-point reply to their comments and suggestions.

Reviewer 1 :

In the manuscript, the authors provide an overview of the various mechanisms by which animal cells sense and respond to viral pathogens. The authors do a satisfactory job at discussing the main components of innate immune sensing and signal transduction.

Where I found that the manuscript fell short is in integrating the description of the sensing/signaling components into a coherent discussion of the greater evolutionary context of their evolution. Because of this, there was a disconnect between the title and abstract, which seems to promise an integrated evolutionary discussion of innate immunity, and the body of the text which in many places has a strong bent towards mammalian systems/viruses. The manuscript would benefit greatly if the evolutionary picture of the systems being discussed were better integrated into the text.

Major Comments

'Rnai v/s PRR-based antiviral immunity' (Lines 395-424): As noted in the manuscript, the evolution of type I IFN appears to have been a defining split in vertebrate evolution. Many/most of the audience of this review will be familiar with IFN-I and its effectors, but will be less familiar with innate immunity in organisms lacking this core component of the mammalian innate immune response. Thus, in my opinion, it would make sense to elucidate this crucial difference very early on in the manuscript – perhaps as part of an expanded 'Vertebrate antiviral immunity' section.

We moved the RNAi section up in the manuscript (after introduction). We agree that it makes more sense to start by describing this system that predated the emergence of adaptive and PRR based immunity.

'Defense & counter defenses' section (Lines 426-430): It would be nice to provide some specificity to the discussion of strategies viruses use in evading the innate immune response. The vast majority of viruses encode one or more proteins whose function it is to counteract or modify the activity of key proteins involved in the innate immune response. Whole reviews have been written on this topic, so clearly it would be impossible for the authors to cover this in depth. However, as written the audience is left in the dark as to how viruses are able to side-step the innate immune response to successfully replicate in their hosts.

As suggested, we expanded the defense & counter defenses section now, giving precise examples of virus evasion strategies from TLR, RLR and cGAS-STING signaling.

Minor Comments

Lines 16/17: Humans should be included in the list of organisms from which our knowledge of innate immune system is derived.

Done

Line 32: This sentence seems to overstate the frequency of virus-host interactions that are beneficial to the host.

‘Many cases’ have been replaced by ‘some cases’.

Line 50: I would suggest replacing 'humans' with 'vertebrates', because, as noted later in the paragraph, humans are not the only species that utilize IFN.

Done

Lines 64/65: This sentence is awkwardly worded.

We have reworded this sentence to make it more clear.

Figure 2: As a means of providing a more holistic overview of immune system evolution, it would be nice to include columns for specialized phagocytic cells and adaptive immunity.

Unfortunately due to space limitations we could not add additional aspects on phagocytotic cells, which we feel is beyond the scope of our article.

Lines 86-88: The adaptive immune system described here applies to vertebrate species from jawed fish to mammals. However, a similar adaptive immune system independently evolved in jawless fish. It would be nice to note this and highlight how much of the underlying signaling is conserved between T cells and VLR-A cells as well as B cells and VLR-B cells.

The adaptive immune system in jawless vertebrates is now highlighted as suggested.

Line 99: The acronyms PAMP and DAMP should be properly defined (pathogen-associated molecular pattern and danger-associated molecular pattern, respectively).

It is now the case

'RNA viruses' section (Lines 133-149): It would be nice to note somewhere in this section that all RNA viruses produce dsRNAs as part of their replication cycle, so both ssRNA and dsRNA viruses have the potential to be sensed by MDA5/TLR3.

This is now noted

Line 258: Rephrase as 'all over the board' does not have any biological/evolutionary meaning.

Done

Lines 266-267: Genes with a history of strong positive selective pressures often have expanded copy numbers and a high prevalence of pseudogenes (see e.g. olfactory receptors and certain tripartite motif-containing genes).

This has been rectified

Lines 267-269: It is unclear what the authors are proposing with this sentence

This sentence have been deleted.

Reviewer 2 Report

Excellent and very readable review. Unique in its effort trying to integrate recent evolutionary evidence into the vast knowledge on innate immunity from the mammalian system. Also this perspective clearly helps understanding the latter.

Very comprehensive list of literature reviewed with helpful combination of review and original articles.

Minor comments:

L42: FIG-1 is obviously very busy. Would it be possible to increase its size and/or resolution?

L130: I suggest to insert after 'sensors' <RLRs and cGAS> to be more explicit

L141: Somewhat contradictory statement. Though DENV and WNV have been shown to activate both RIG-I and MDA5, their RNA do not possess free 5' di- or triphosphates but a canonic cap. Many be a somewhat misleading example for people outside of the field.

Some minor typos:

L98: pathogenic (delete 'ic') molecular patterns

L252: Trichoplax in Italics

L339+342: Vago should be in bold to be consistent with other proteins/genes mentioned

L361: could not (instead of couldn't)

L377: specie<s>

L429: Plant viruses (no plural-s in plant)

Throughout the text: extra spaces or spaces missing (e.g. L122 before referencing)

Author Response

We would like to thank the reviewers for their thorough reading of the manuscript. Their comments were very insightful and we took into account all their suggestions in the revised manuscript. Please find below a point-by-point reply to their comments and suggestions.

Reviewer 2

Comments and Suggestions for Authors

Excellent and very readable review. Unique in its effort trying to integrate recent evolutionary evidence into the vast knowledge on innate immunity from the mammalian system. Also this perspective clearly helps understanding the latter.

Very comprehensive list of literature reviewed with helpful combination of review and original articles.

Minor comments:

L42: FIG-1 is obviously very busy. Would it be possible to increase its size and/or resolution?

In Fig1 we tried to generate a comprehensive overview of the reviewed signaling pathways. To make sure the resolution of the figure guarantees readability, we will provide both, a high resolution figure as well as the original vector graphic in order to allow free scaling of the figure.

L130: I suggest to insert after 'sensors' <RLRs and cGAS> to be more explicit

Done

L141: Somewhat contradictory statement. Though DENV and WNV have been shown to activate both RIG-I and MDA5, their RNA do not possess free 5' di- or triphosphates but a canonic cap. Many be a somewhat misleading example for people outside of the field.

It has been shown that although WNV and DENV use a canonical cap-dependent translation and their incoming virions lack phosphsorylated or dsRNAs, WNV and DENV infected cells produce RNAs containing 5′ triphosphate and double-stranded RNA that are temporally distributed and sensed by RIG-I and MDA5 during infection (PMID : 23966395 , 17942531 ).

Some minor typos:

L98: pathogenic (delete 'ic') molecular patterns

Ok- corrected.

L252: Trichoplax in Italics

Ok- corrected.

L339+342: Vago should be in bold to be consistent with other proteins/genes mentioned

Ok- corrected.

L361: could not (instead of couldn't)

Ok- corrected.

L377: specie<s>

Ok- corrected.

L429: Plant viruses (no plural-s in plant)

Ok- corrected.

Throughout the text: extra spaces or spaces missing (e.g. L122 before referencing

Ok- corrected and the rest of the text have been double checked.